# Thermogravimetric Experiment of Urea at Constant Temperatures

**DOI:** 10.3390/ma14206190

**Published:** 2021-10-18

**Authors:** Neng Zhu, Feng Qian, Xiaowei Xu, Mingda Wang, Qi Teng

**Affiliations:** 1School of Automotive and Transportation Engineering, Wuhan University of Science and Technology, Wuhan 430081, China; znqc@wust.edu.cn (N.Z.); Xiaowei.xu@wust.edu.cn (X.X.); 2Chinese Academy of Environmental Sciences, Beijing 100012, China; wangmingda@vecc-mee.org.cn (M.W.); tengq@vecc.org.cn (Q.T.)

**Keywords:** thermal decomposition, deposits, selective catalytic reduction, diesel engine

## Abstract

There are still many unsolved mysteries in the thermal decomposition process of urea. This paper studied the thermal decomposition process of urea at constant temperatures by the thermal gravimetric–mass spectrometry analysis method. The results show that there are three obvious stages of mass loss during the thermal decomposition process of urea, which is closely related to the temperature. When the temperature was below 160 °C, urea decomposition almost did not occur, and molten urea evaporated slowly. When the temperature was between 180 and 200 °C, the content of biuret, one of the by-products in the thermal decomposition of urea, reached a maximum. When the temperature was higher than 200 °C, the first stage of mass loss was completed quickly, and urea and biuret rapidly broke down. When the temperature was about 240 °C, there were rarely urea and biuret in residual substance; however, the content of cyanuric acid was still rising. When the temperature was higher than 280°C, there was a second stage of mass loss. In the second stage of mass loss, when the temperature was higher than 330 °C, mass decreased rapidly, which was mainly due to the decomposition of cyanuric acid. When the temperature was higher than 380 °C, the third stage of mass loss occurred. However, when the temperature was higher than 400 °C, and after continuous heating was applied for a sufficiently long time, the residual mass was reduced to almost zero eventually.

## 1. Introduction

Interest in the study of urea has continued to grow since urea was found by Roselle [1] in 1773. In 1828, German chemist Wohler [2] converted inorganic ammonium cyanate to organic urea by heating method for the first time, which, for the first time, broke the traditional idea that organics can only be obtained from organic compounds. The synthesis of urea has opened a prologue of artificial synthesis organics, and it is thought to be a pioneer in organic chemistry research. Nowadays, urea has been widely used in many fields such as the medical industry, agriculture, manufacture, and commerce and is closely related to human production and living. Even so, there are still many mysteries in the thermal decomposition process of urea.

At present, selective catalytic reduction (SCR), which uses urea water solution (UWS) as the reductant, is widely regarded as one of the most promising technologies to reduce NO*_x_* emissions from diesel engines and to meet increasingly stringent emission regulation standards [3,4,5]. However, there are many problems derived from the urea decomposition process. One of them is that deposit formation on catalyst and inlet pipe surfaces blocks the pores and the active sites of catalyst, which decreases the durability of the SCR systems. Therefore, a better understanding of the urea decomposition process can lead to improvement in the SCR performance and overcome this shortcoming [6].

It is well known that urea starts to break down at about 150 °C and generates equimolar NH_3_ and HNCO (reaction (1)) [7,8,9,10]. HNCO is prone to hydrolysis (reaction (2)). However, in the process of urea pyrolysis, there will be multiple side reactions, and the generated by-products are closely related to the temperature. Schaber’s research [11] showed that when the temperature was above 152 °C, the decomposition of urea occurred (reaction (1)). At about 160 °C, HNCO, the product of urea pyrolysis, reacted with the remaining urea to generate biuret (reaction (3)). At about 175 °C, cyanuric acid (reactions (4) and (5)) and ammelide (reaction (6)) began to generate. At about 250 °C, the ammeline started to generate (reaction (7)). Thagard [12] dealt with the chemical analysis of the by-products formed by urea decomposition at about 150–200 °C in a dielectric barrier discharge (DBD). Recent studies [13,14,15,16,17] showed that we could obtain hydrogen by urea pyrolysis reaction, which was considered as a potential source of hydrogen/fuel cell power. Jenny [17] studied the thermal decomposition of urea and aqueous urea solutions in the presence of 18 wt% nickel on alumina. Steffen’s research [18] observed and explained the phenomenon of liquefaction and resolidification of biuret in the temperature range of 193~230 °C. In addition, Stradella [19], Carp [20], and Lundstrom [21] have also carried out related studies on urea pyrolysis. Furthermore, the kinetic analyses of the thermal decomposition reaction of urea and related by-product can be evaluated using thermogravimetric analysis [22,23]. However, in these studies, the thermal decomposition process of urea under constant temperatures condition is not described.
Urea pyrolysis: (H_2_N)_2_CO → NH_3_ + HNCO(1)
HNCO hydrolysis: HNCO + H_2_O → NH_3_ + CO_2_(2)
Generation of biuret: (H_2_N)_2_CO + HNCO → (H_2_NCO)_2_NH(3)
Generation of cyanuric acid: (H_2_NCO)_2_NH + HNCO → CYA + NH_3_(4)
Generation of cyanuric acid: 3HNCO → CYA(5)
Generation of ammelide: (HNCO)_3_ + NH_3_ → ammelide + H_2_O(6)
Generation of ammeline: ammelide + NH_3_ → ammeline + H_2_O(7)

SCR is a technology to deal with NO*_x_* emission from diesel engines, and its reducing agent is ammonia gas produced by the decomposition of urea. In this process, undesired solid by-products, commonly known as deposits, are formed in the exhaust pipe. Deposits, in general, show in three places of the SCR system: interior part of urea injector, exhaust pipe wall, and catalyst entrance. Deposits on the exhaust pipe wall may decrease the utilization rate of urea. Concentration weakening of NH_3_, which reacts with SCR, could lead to reducing the conversion efficiency of the SCR system. A considerable quantity of deposits may cause partial and even complete blockage of the exhaust pipe, increase exhaust backpressure, and thus influence the performance of the engine. Hence, in order to guarantee the normal operation of the SCR system and engine, addressing deposit-related problems on the exhaust pipe wall is very urgent.

In order to explore the underlying causes and influential factors in deposits’ generation, scholars carried out numerous relevant studies [24,25]. Way [26] analyzed the composition of deposits. The outcome indicates deposits are composed of undecomposed urea (occupy 5~15%), biuret (occupy 5~15%), and CYA (occupy 70~95%). Research by Zheng [27] indicates pipe wall temperature exerts a certain influence on the formation of deposits. Strots [28] conducted a study on the proportion of deposits in urea consumption under different exhaust temperatures and ambient temperatures, the results of which showed that the lower the exhaust temperature is, the larger the proportion of deposits will account for. Xu [29] researched the influence of exhaust temperature and catalyst types (iron zeolite and copper zeolite) on deposits production, the results of which showed that the lower the exhaust temperature is, the more easily the deposits form.

The formation and decomposition of these by-products have a strong correlation with temperature. This was our motivation to systematically investigate the decomposition of urea at constant temperatures by performing TG experiments. Therefore, in this study, the thermal decomposition process of urea with continuous heating under constant temperatures was investigated utilizing the thermogravimetric (TG) analysis method, in parallel with detecting the evolved gases by mass spectrometry (MS). This research helps to solve the problem of deposits in diesel SCR systems.

## 2. Materials and Methods

The urea used in the experiments was produced by Tianjin Guangfu Technology Development Co. Ltd., and the purity was not less than 99%. Urea was heated by a STA449F3 model synchronized thermal analyzer, produced by NETZSCH. In the process of heating, temperatures were controlled by programs, and urea mass loss could be measured. Usually, approximately 10 mg samples of urea is put on the alumina crucible. Argon was the purging gas in the heating furnace, and the purging velocity was 40 mL/min. In addition, a QMS403C model quadrupole mass-spectrum analyzer was also used synchronously with the thermal analyzer to detect the escaping gas composition in the process of thermal decomposition of urea.

In this article, urea was heated from ambient temperature to 1000 °C with a heating rate of 10 °C per minute. The thermogravimetric curve is shown in Figure 1. In addition, urea was also heated from ambient temperature to different target temperatures, with the fastest heating rate of 40 °C per minute, then the apparatus maintained a constant temperature for a period of time until the residual mass was no longer change. Specifically, the constant temperatures were kept for 3 h when target temperatures were 140 °C, 160 °C, and 280 °C, 1 h when target temperatures were 180 °C, 200 °C, and 240 °C, and 40 min when target temperatures were 320 °C, 360 °C, and 400 °C. At different target temperatures, residual masses of temperature rising experiment and constant temperature experiment are shown in Table 1.

## 3. Results

### 3.1. Temperature Rising Thermogravimetric Experiment

Figure 1 shows the TG-MS results of urea in the temperature-rising experiment. From the thermogravimetric curve, we can infer that there were three obvious stages of mass loss in the urea pyrolysis process. The first stage of mass loss was between 140 °C to 250 °C, and there was about 68.6% mass loss. The second stage concluded at about 360 °C and about 26.7% mass loss. The third concluded at about 410 °C and about 3.6% mass loss.

Based on mass spectrum curves in Figure 1, when the temperature reached urea melting point 133 °C, the ion current intensity of evolved gases, NH_3_ and HNCO, increased slowly. This is due to the fact that after evaporation, molten urea decomposed in the process of entering the mass spectrograph through a capillary whose temperature was 280 °C (reaction (1)). When the temperature was higher than 152 °C, the ion current intensity of NH_3_ and HNCO began to increase noticeably, which suggested that urea started to decompose. Until about 230 °C, the concentration of NH_3_ and HNCO was observed reaching a maximum value almost at the same time. It is obvious that the concentration of HNCO was far less than NH_3_, so it can be inferred that the generated HNCO was consumed partially through other reactions. Schaber [11] and Kleemann [30] also observed this phenomenon. At the same time, CO_2_ was observed to escape, which indicated that the hydrolysis reaction of HNCO might happen (reaction (2)). Kleemann’s [30] research shows that HNCO is a very stable gas, and the hydrolysis reaction more likely occurs on the catalyst surface, so there are other reactions for the consumption of HNCO. When the temperature was higher than 160 °C, residual urea would react with HNCO to generate biuret (reaction (3)), and the biuret would react with HNCO to generate CYA (reaction (4)). However, the thermogravimetric curve of biuret showed that at about 190 °C, biuret decomposed (reaction (8)) to generate HNCO; therefore, we can infer that there must be other reactions for the consumption of HNCO. Based on the studies of our predecessors, the polymerization of HNCO to generate CYA is considered a reasonable reaction to explain the above phenomenon (reaction (5)). The ion current intensity of H_2_O increased because of the occurrence of reactions 6 and 7. When the temperature reached about 265 °C, the low concentration of NH_3_ and HNCO indicated that there were little urea and biuret in the residual material. Until 330 °C, the variation trend of the thermogravimetric curve was flat, and the concentration of NH_3_ and HNCO was still relatively low. When the temperature was between 330 and 360 °C, the concentration of HNCO reached another peak value, which was caused by the rapid decomposition of cyanuric acid (reaction (9)). When the temperature was above 360 °C, cyanuric acid decomposed continually, and there was still a small amount of HNCO that continued to escape. From then on, ammelide, ammeline, and other by-products decomposed one after another.
Decomposition of biuret: (H_2_NCO)_2_NH → (H_2_N)_2_CO + HNCO(8)
Decomposition of cyanuric acid: CYA → 3HNCO(9)

### 3.2. Thermogravimetric Experiments at Constant Temperatures

Given that the pyrolysis process of urea has a close relationship with temperature, in this paper, nine target temperatures between 140 and 400 °C were selected to carry out the thermogravimetric experiments.

When the targeted temperature was 140 °C (Figure 2), urea mass decreased slowly. As the temperature was not enough high for the decomposition of urea, the slow evaporation of molten urea was the reason for mass loss. As can be seen from the mass spectrum curve, a small amount of NH_3_ and HNCO was observed to escape, which was because after evaporation, molten urea decomposed in the process of entering the mass spectrograph through a capillary whose temperature was 280 °C. After 180 min of continuous heating, urea mass was no longer reduced, and the final residual mass accounted for 26.87% of the total mass. Urea mass did not decrease to zero eventually, which indicated that under this temperature, urea would slowly turn to be other substances.

When the targeted temperature was 160 °C (Figure 3), urea mass still declined slowly. Although it reached the urea pyrolysis temperature, the ion current intensity of NH_3_ was still low and close to the case of 140 °C. The above phenomenon indicated that the urea pyrolysis rate was slow in this case, and evaporation of urea was still the main reason for weight loss. In addition, we can see that the ion current intensity of HNCO in the case of 160 °C was lower than that of 140 °C, which suggested that when the targeted temperature was 160 °C, the polymerization reaction of HNCO occurred. After heating continuously for 80 min, urea mass was no longer reduced, and the final residual mass accounted for 35.32% of the total mass.

When the targeted temperature was 180 °C (Figure 4), the variation trend of urea mass was similar to that in the former cases. The only difference was that the required time to reach a steady state was shortened. Observed from the result of mass spectrometry, ion current intensity of NH_3_ and HNCO significantly increased, which indicated that urea pyrolysis at this temperature was the main reason for weight loss. After heating continuously for 25 min, remnants reached a steady state, and the final residual mass accounted for 36.53% of the total mass.

When the targeted temperature was 200 °C (Figure 5), there were two stages of mass loss in the urea pyrolysis process. The first stage lasted for about 5 min. There was a large amount of NH_3_ and HNCO escaping and residual mass, which accounted for 47.5% of the total mass. This mass loss was mainly caused by urea pyrolysis. In the second stage, residual mass declined slowly. After continuously heating for 30 min, mass loss was about 6.5%. The melting point of biuret is 190 °C, and its thermal decomposition products are urea and HNCO (Equation (8)). From the mass spectrum curve, it can be observed that the concentration of NH_3_ and HNCO was extremely low; thus, the evaporation of biuret was the main cause of mass loss in the second stage.

Based on the above analyses, we can infer that when the temperature was below 200 °C, with temperature increasing, the duration of the first mass loss stage in the urea pyrolysis process became shorter, and the final residual mass increased. Between 140 and 160 °C, urea pyrolysis hardly occurred, and evaporation of molten urea was the main feature in this temperature range. Between 160 and 180 °C, urea started to decompose, and the higher the temperature was, the shorter was the time to reach a steady state (i.e., residual mass no longer changed), and the faster was the urea pyrolysis rate. Even though the urea pyrolysis rate was higher in the case of 180 °C, the residual mass was still about 1% higher than the case of 160 °C, which suggested that the formation rate of by-products also increased. Between 180 and 200 °C, urea decomposed quickly, and the first stage of mass loss was completed in a short time. However, at 200 °C, the slow evaporation of biuret led to the second stage of mass loss, so the amount of biuret in the residual material reached a maximum in the temperature range of 180~200 °C.

When the targeted temperature was 240 °C (Figure 6), there was only one stage of mass loss, and the escaping amount of NH_3_ and HNCO increased rapidly. Hence, a large amount of biuret decomposed at this temperature. The process of mass loss was accomplished within 5 min, which suggested that the biuret pyrolysis rate was high. Finally, residual mass accounted for 28.3% of the total mass. There was little urea and biuret in the residual material, and the main ingredient was cyanuric acid.

When the targeted temperature was 280 °C (Figure 7), the two stages of mass loss were similar to the case of 200 °C. The first stage of mass loss was completed in the process of rapid temperature rising, and the residual mass decreased from 100% to 36% of the total mass within 4 min. Mass spectrometry results showed that a large amount of NH_3_ and HNCO escaped. The main ingredient was cyanuric acid in residual material at the end of the first stage. The second stage of mass loss lasted for about 70 min, and the mass loss was about 25%. Based on the mass spectrometry results, HNCO was not visibly detected, which suggested that under this temperature, cyanuric acid could hardly decompose. The second stage of mass loss was mainly caused by the sublimation of cyanuric acid and other by-products. Even at 280 °C, after a sufficiently long time of heating, the residual mass still did not reach zero, and the final residual mass accounted for 8.1% of the total mass.

Based on the above analyses, we can see that when the temperature was above 240 °C, there was little urea and biuret in the residual material, and the main ingredient in the residual material was cyanuric acid. Compared with the case of 240 °C, it was obvious that the residual mass at the end of the first stage was more in the case of 280 °C, which indicated that the generation of cyanuric acid increased in the temperature range of 240~280 °C.

When the targeted temperature was 320 °C (Figure 8), 360 °C (Figure 9), and 400 °C (Figure 10), as can be observed, there were two stages of rapid mass loss on the thermogravimetric curve. The first stages of mass loss were all completed before 280 °C, and the heating rate was high enough; therefore, in the three experiments, thermogravimetric curves in the first stages of mass loss basically coincided. At the end of the first stage of mass loss, the residual mass in three experiments, respectively, accounted for 30.43%, 33.35%, and 33.19% of the total mass. As can be seen from the results of mass spectrometry, in all these three experiments, there was a large amount of NH_3_ and HNCO escaping in the first stage of mass loss. The escaping amount was almost the same, and the peak value appeared near 250 °C.

Based on the above conclusions, at the end of the first stage of mass loss, almost no urea and biuret were contained in residual material, and the main ingredient was cyanuric acid. Therefore, the second stage of mass loss was mainly the decomposition process of cyanuric acid.

Compared with the cases of 360 °C and 400 °C, when the constant temperature was 320 °C, the thermogravimetric curve decreased relatively gently in the second stage of mass loss, and there was also an inconspicuous emergence of HNCO escaping observed from the mass spectrometry results, which indicated that cyanuric acid pyrolysis rate was low. When the constant temperatures were 360 °C and 400 °C, an obvious mass loss was observed, and HNCO started to escape when the temperature reached 330 °C. Therefore, when the temperature was higher than 330 °C, cyanuric acid began to decompose rapidly.

In addition, when the constant temperature was 400 °C, a third stage of mass loss appeared. As can be seen from the thermogravimetric curve, the third stage of mass loss began when the temperature was higher than 380 °C, and the final residual mass was almost zero (accounted for 0.14% of the total mass).

## 4. Discussion

The residual mass is closely related to the heating rate. When achieving the same targeted temperature, the slower the heating rate is, the longer is the continuous heating time of the urea sample, and the more complete is the thermal decomposition process, resulting in less residual mass. As shown in this paper, the pyrolysis process of urea at constant temperature for a long enough time was described in detail. The experimental results of urea-related deposit in the SCR system of diesel engines were explained from the perspective of the pyrolysis mechanism. Furthermore, research in the literature was supported.

When the SCR system operated below 200 °C for a long time, it was prone to form deposits such as biuret and lead to reducing the conversion efficiency of the SCR system. We could infer that the NO*_x_* conversion efficiency was no more than 65% through the residual mass. Strots’ research [28] showed that it was likely to form deposits when the exhaust temperature was lower than 200 °C. Under the above condition, there were about 25~65 wt% of urea converting to the deposits.

When the SCR system operated below 300 °C, there was still a risk of forming deposits of CYA, which was more difficult to remove. Xu’s research [29] showed that there were deposits formed, and the whole catalyst was almost blocked when the temperature of the SCR catalyst was 250 °C.

When the SCR system operated above 330 °C for a period of time, the existing deposits were able to be eliminated. Strots’ research showed that there was only less than 1 wt% of urea converting to the deposits when the exhaust temperature was achieving 350 °C. Xu’s research showed that there was no deposit forming on the catalyst when the temperature of the SCR catalyst was 350 °C.

To summarize, the results of this paper could provide a theoretical basis to solve the problem of urea-related deposits for the SCR system.

## 5. Conclusions

The urea pyrolysis process with the temperature rising at a certain heating rate has been studied by many researchers. In this paper, a temperature-rising thermogravimetric experiment of urea was carried out, and the mass loss of urea under different temperatures was observed to be similar to the results of predecessors. The urea pyrolysis process has a close relationship with the temperature, so thermogravimetric experiments of urea at constant temperatures were also studied in this paper.

When the targeted temperature was between 140 and 180 °C, urea mass reduced very slowly. If the heating rate was high enough, there would barely be any mass loss. If urea remained in the range of 140~180 °C for a long time, the final residual mass accounted for about 27~37% of the total mass. In addition, when the targeted temperature was between 140 and 160 °C, urea hardly decomposed, so evaporation of molten urea was the main reason for mass loss. When the targeted temperature was between 160 and 180 °C, urea began to decompose slowly. With the temperature increasing, the urea pyrolysis rate increased gradually, and the generation of by-products also increased at the same time.

When the targeted temperature was between 180 and 200 °C, urea began to decompose rapidly, and the amount of biuret in the residual material reached its maximum when the temperature was in the range of 180~200 °C. When the targeted temperature was between 200 and 240 °C, biuret began to decompose rapidly. If urea stayed in this temperature for enough long time, the residual mass finally accounted for about 28~39% of the total mass.

When the targeted temperature was between 240 and 280 °C, there was a rapid decline in mass during the first stage, and this process was completed within 5 min. The amount of cyanuric acid in the residual material gradually rose, and little urea and biuret were contained in the residual material. After keeping the temperature for a sufficiently long time, eventually, the residual by-products accounted for about 9~28% of the total mass.

When the targeted temperature was between 320 and 400 °C, after the first stage of mass loss, residual mass accounted for about 30% of the total mass. After keeping the temperature for a sufficiently long time, residual material was less than 7% of the total mass in this temperature range.

The urea pyrolysis process has three obvious stages of mass loss. When the temperature was above 200 °C, the first stage of mass loss was completed quickly and basically finished before the temperature reached 280 °C. When the targeted temperature was above 330 °C, there was a rapid decline in mass during the second stage of mass loss. When the temperature was above 380 °C, there was a third stage of mass loss. When the temperature was above 400 °C, residual mass eventually decreased to almost zero after continuous heating for a sufficiently long time.

## Figures and Tables

**Figure 1 materials-14-06190-f001:**
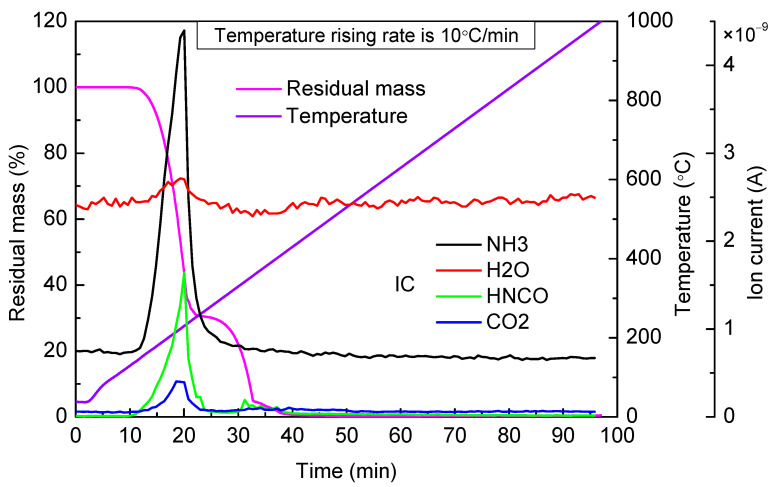
TG-MS results of urea in temperature-rising experiment.

**Figure 2 materials-14-06190-f002:**
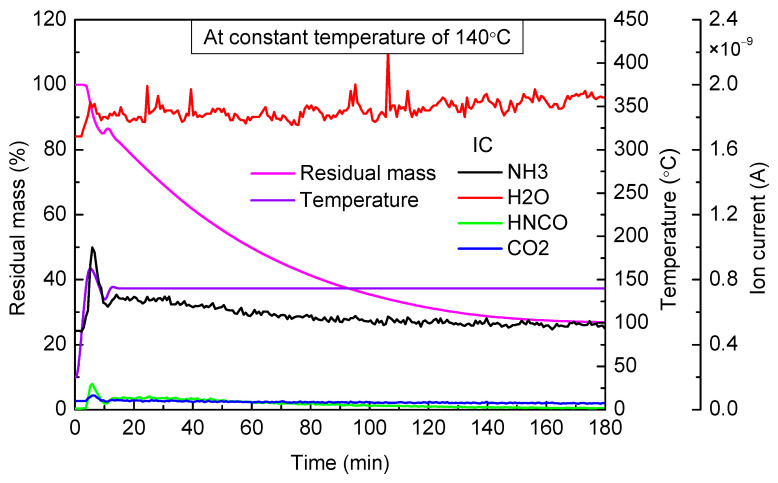
TG-MS results of urea at constant temperature of 140 °C.

**Figure 3 materials-14-06190-f003:**
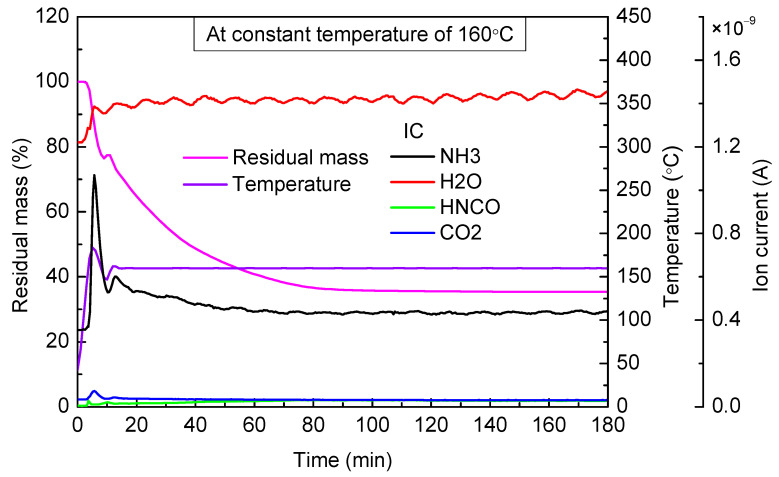
TG-MS results of urea at constant temperature of 160 °C.

**Figure 4 materials-14-06190-f004:**
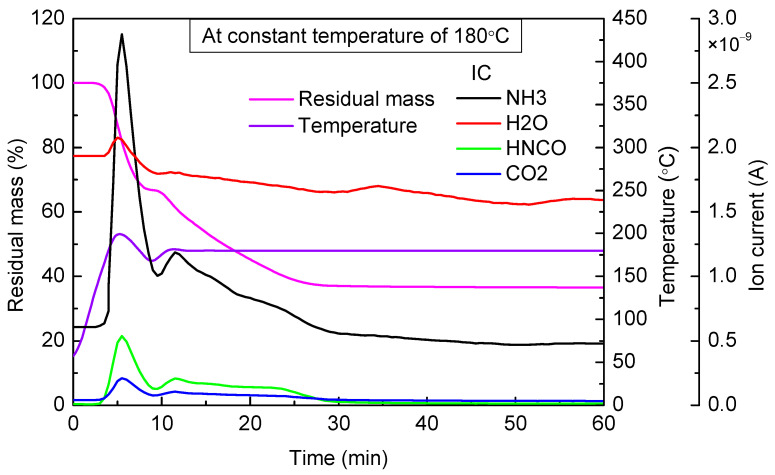
TG-MS results of urea at constant temperature of 180 °C.

**Figure 5 materials-14-06190-f005:**
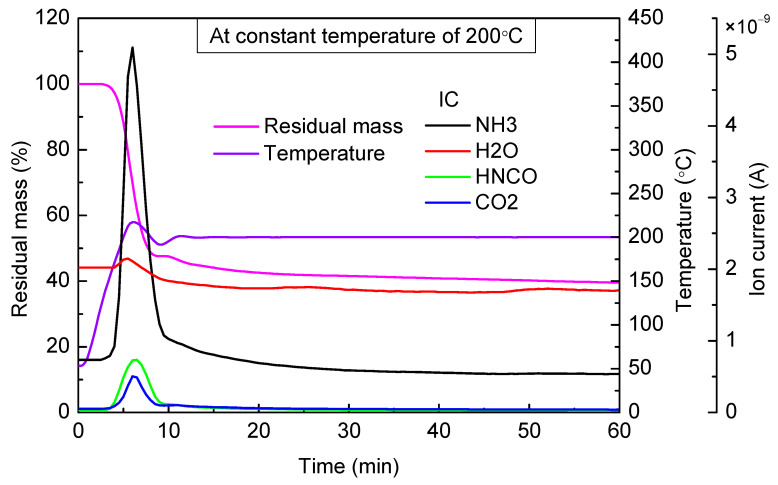
TG-MS results of urea at constant temperature of 200 °C.

**Figure 6 materials-14-06190-f006:**
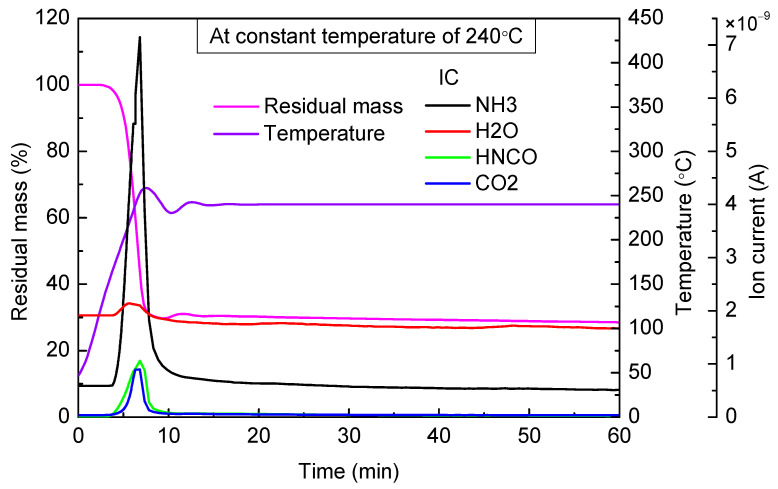
TG-MS results of urea at constant temperature of 240 °C.

**Figure 7 materials-14-06190-f007:**
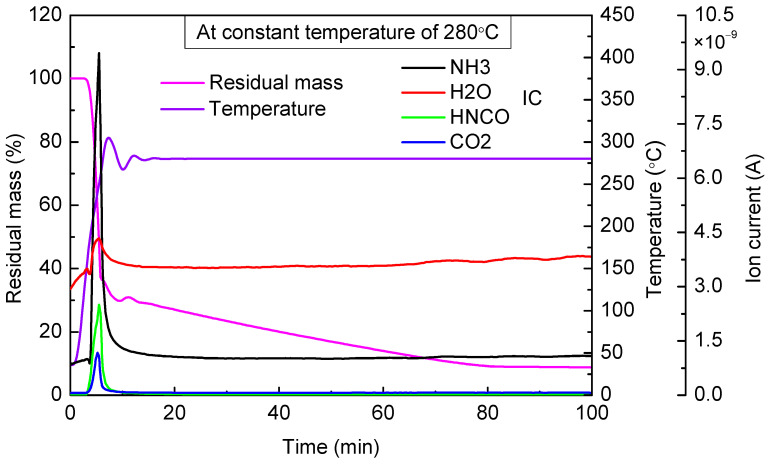
TG-MS results of urea at constant temperature of 280 °C.

**Figure 8 materials-14-06190-f008:**
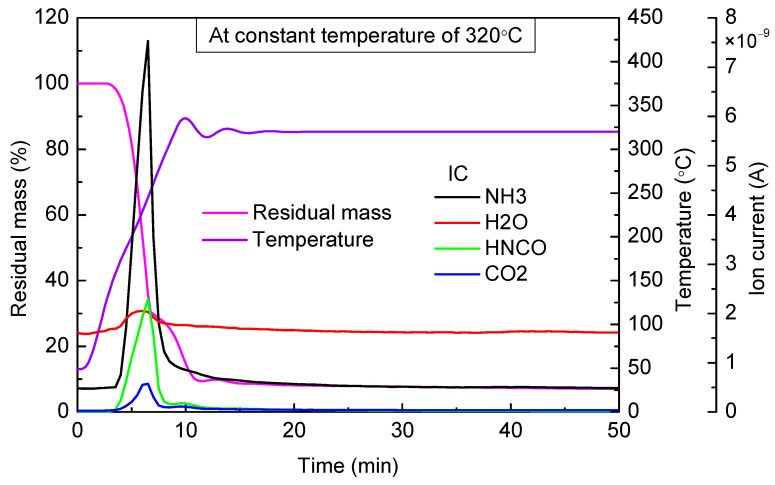
TG-MS results of urea at constant temperature of 320 °C.

**Figure 9 materials-14-06190-f009:**
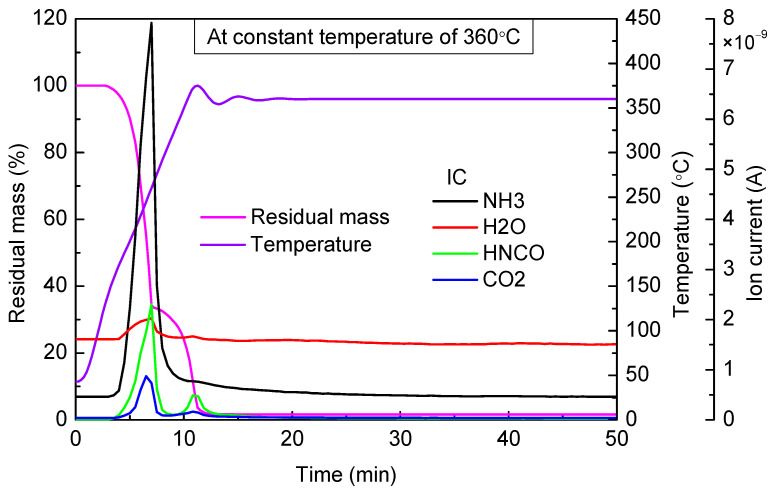
TG-MS results of urea at constant temperature of 360 °C.

**Figure 10 materials-14-06190-f010:**
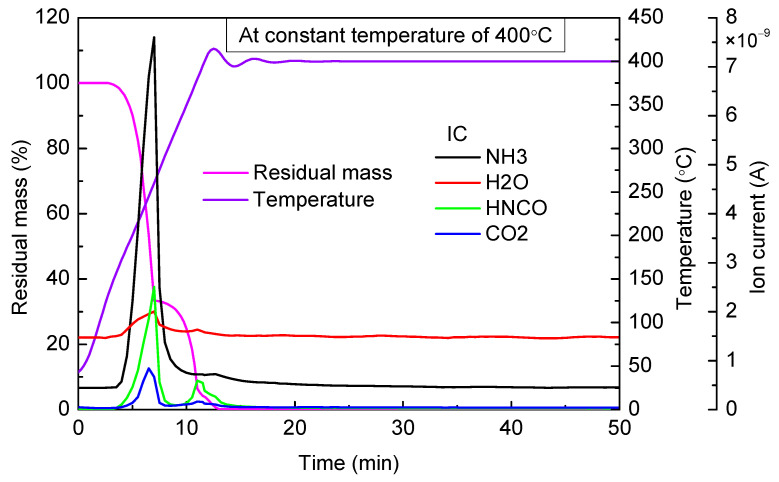
TG-MS results of urea at constant temperature of 400 °C.

**Table 1 materials-14-06190-t001:** Residual mass corresponding to different target temperatures.

Target Temp. (°C)	Residual Mass (%)		
	Temp. Rising Expt.	Constant Temp. Expt.	
		Initial Mass	Final Mass
140	99.79	99.66	26.87
160	97.64	97.42	35.32
180	90.99	96.74	36.53
200	78.05	88.34	39.10
240	34.22	58.60	28.35
280	30.09	37.17	8.84
320	25.18	22.85	7.12
360	4.70	19.65	1.65
400	1.97	3.74	0.14

## Data Availability

Not applicable.

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
