# Peer review of "Thermogravimetric Experiment of Urea at Constant Temperatures"

_materials, 2021, doi:10.3390/ma14206190_

Round 1
Reviewer 1 Report
The manuscript entitled “Research on Thermo-gravimetric Experiment of Urea at Constant Temperatures” submitted by Feng Qian and co-workers for consideration for publication in the MDPI journal Materials presents the investigation of the thermal decomposition process of urea at constant temperatures by the thermal gravimetric - mass spectrometry analysis method. The manuscript is well written, and the results are well discussed. However, I do not understand why the authors did not show in the manuscript the following recently published paper: “Thermodynamics and reaction mechanism of urea decomposition”, Phys. Chem. Chem. Phys., 2019, 21, 16785.
What is the novelty of the presented manuscript in comparison to already published papers? I recommend conducting a comprehensive literature review and deep description of the novelty of the presented work. What are the reasons that this article will be of interest and importance to other researchers? What is the novelty of the presented data? How can the other scientists use them?
There are other important papers such as: Thermochimica Acta 565 (2013) 39-45, Thermogravimetric evolved gas analysis of urea and urea solutions with nickelalumina catalyst.
I recommend at least a major revision of the manuscript.
Reviewer 2 Report
Title: Research on Thermo-gravimetric Experiment of Urea at Constant Temperatures
Manuscript ID: materials-1402343
Authors: Zhu et al.
Dear Authors,
Thank you for the opportunity to read your article. I found the topic is interesting. Generally speaking, there are some results presented in order to capture some trends. On the other hand, the results need more deeper discussion and the summary of your findings needs to be provided. I suggest that this article will be revised before resubmission for another review process. As a conclusion, I recommend its major revision at this state.
I hope my comments are helpful.
Good luck,
A reviewer
Major concerns:
Article title:
-Please consider removing “Research on” from the title since any article reports the results of research.
“Keywords”
-Please consider listing keywords that are not used in the article title.
“3. Results”, “4. Discussion”
-There are many results presented, but they are simply described without deep discussion to extract generalities. The contents stated in “4. Discussion” are rather descriptive but not actual discussion based on the results and literature. Please try to extract more general essence from your results and cite relevant values from literature for comparison. In other words, please try to include interpretations, implications, limitations and recommendations.
“Conclusions”
-You may create a section called Conclusions and summarize your findings there.
-You may state some future perspectives.
Reviewer 3 Report
In the manuscript entitled “Research on Thermo-gravimetric Experiment of Urea at Constant Temperatures” authors present a study on thermal decomposition process of urea using thermal gravimetric - mass spectrometry analysis method.
The article could be important for technical and technological applications. However, some deficiencies exist, which are listed below.
- In the introduction I’m missing reference and discussion of major work on kinetic study of thermal decompositions of functional materials, no reference to this substantial contribution to this area is given but furthermore, that research should be discussed in these context as it is highly relevant, for example: J Therm Anal Calorim. 2017, 130, 799–812, Organomet. Chem. 2017, 847, 173–183, but many others are available.
- TGA has not been sufficiently analyzed. There is no kinetic study.
- The manuscript needs a correction, in many places there are editing errors. For example no superscripts in Figures 1-10.
- The Conclusions section should be improved. There is no clear explanation (for a wide audience) of significant novelty of the work as compared with their previous work. This should be highlighted and compared with their previous work. What are the new findings in this work and how it differs from their previous work?
Round 2
Reviewer 1 Report
The revised manuscript entitled “Thermo-gravimetric Experiment of Urea at Constant Temperatures” submitted by Feng Qian and co-workers for reconsideration for publication in the MDPI journal Materials presents a bit higher level than the first submission. The Authors have performed the required corrections partially and add additional explanations. Nevertheless, there are still elements that should be revised.
- Twenty-one literature positions were changed on 26, so 5 references were added, but there are still other positions that, in my opinion, should be added. The authors are aware that the subject which they present is already comprehensively described in the literature, so a detailed study of the literature is necessary. As I can see, other reviewers also flag up a problem.
- The authors’ list is already changed, two additional co-authors were added. Could You explain it?
Therefore, I consider the revised manuscript may be suitable for publication in the MDPI journal Materials after the next revision. I recommend the major revision.
Reviewer 2 Report
Dear Authors,
Thank you for your efforts to address the comments and concerns. Now, I see the clarity of the article has been significantly improved and would suggest that the journal accepts it for publication.
Best regards,
A reviewer
Reviewer 3 Report
Authors improved their manuscript and answered my questions / suggestions, it is now suitable for publication, However references require editing, e.g. number 22 the authors first name and surname are replaced.
Round 3
Reviewer 1 Report
The revised manuscript entitled “Thermo-gravimetric Experiment of Urea at Constant Temperatures” submitted by Feng Qian and co-workers for reconsideration for publication in the MDPI journal Materials after the second revision can be published in the MDPI journal Materials. I wish all the best to authors and next good papers with comprehensively studied literature ?